# Analysis of Stress Response Genes in Microtuberization of Potato *Solanum tuberosum* L.: Contributions to Osmotic and Combined Abiotic Stress Tolerance

**DOI:** 10.3390/plants13212996

**Published:** 2024-10-26

**Authors:** Lisset Herrera-Isidron, Braulio Uribe-Lopez, Aaron Barraza, José Luis Cabrera-Ponce, Eliana Valencia-Lozano

**Affiliations:** 1Unidad Profesional Interdisciplinaria de Ingeniería Campus Guanajuato (UPIIG), Instituto Politécnico Nacional, Av. Mineral de Valenciana 200, Puerto Interior, Silao de la Victoria 36275, Guanajuato, Mexico; lherrerai@ipn.mx (L.H.-I.); buribel1900@alumno.ipn.mx (B.U.-L.); 2CONAHCYT-Centro de Investigaciones Biológicas del Noreste, SC. Instituto Politécnico Nacional 195, Playa Palo de Santa Rita Sur, La Paz 23096, Baja California Sur, Mexico; abarraza@cibnor.mx; 3Departamento de Ingeniería Genética, Centro de Investigación y de Estudios Avanzados del IPN, Unidad Irapuato, Irapuato 36824, Guanajuato, Mexico

**Keywords:** stress, microtuberization, potato, osmotic, heat–osmotic, cold–osmotic, salt–osmotic, combined-all stresses, introgression, domestication candidate, gene expression

## Abstract

Wild Solanum species have contributed many introgressed genes during domestication into current cultivated potatoes, enhancing their biotic and abiotic stress resistance and facilitating global expansion. Abiotic stress negatively impacts potato physiology and productivity. Understanding the molecular mechanisms regulating tuber development may help solve this global problem. We made a transcriptomic analysis of potato microtuberization under darkness, cytokinins, and osmotic stress conditions. A protein–protein interaction (PPI) network analysis identified 404 genes with high confidence. These genes were involved in important processes like oxidative stress, carbon metabolism, sterol biosynthesis, starch and sucrose metabolism, fatty acid biosynthesis, and nucleosome assembly. From this network, we selected nine ancestral genes along with eight additional stress-related genes. We used qPCR to analyze the expression of the selected genes under osmotic, heat–osmotic, cold–osmotic, salt–osmotic, and combined-stress conditions. The principal component analysis (PCA) revealed that 60.61% of the genes analyzed were associated with osmotic, cold–osmotic, and heat–osmotic stress. Seven out of ten introgression/domestication genes showed the highest variance in the analysis. The genes *H3.2* and *GAPCP1* were involved in osmotic, cold–osmotic, and heat–osmotic stress. Under combined-all stress, *TPI* and *RPL4* were significant, while in salt–osmotic stress conditions, *ENO1*, *HSP70-8*, and *PER* were significant. This indicates the importance of ancestral genes for potato survival during evolution. The targeted manipulation of these genes could improve combined-stress tolerance in potatoes, providing a genetic basis for enhancing crop resilience.

## 1. Introduction

Potato (*Solanum tuberosum* L.) is the world’s fourth food crop and a pillar of food security. This crop was domesticated between 8000 and 10,000 years ago from wild species in the Andes of southern Peru [1]. In cultivar tetraploid potatoes, wild Solanum species have contributed 407 introgression and 841 domestication candidate genes involved in disease resistance, heat and osmotic stress tolerance, and antioxidant pathways [1]. Introgression (INT) refers to the transfer of genetic material from wild type species into cultivated potatoes through repeated backcrossing. Domestication candidates (DOMc) are genes under selection in both landraces and cultivars and have an impact on performance regardless of hemisphere or local adaptation.

INT genes have improved key traits, such as disease resistance. By the 1980s, six wild species were used in Europe for breeding resistant cultivars against late blight (*S. demissum* and *S. stoloniferum*), viruses (*S. chacoense* and *S. acaule*), and potato cyst nematodes (*S. vernei* and *S. spegazzinii*) [2]. Nevertheless, climate change has a negative impact on potatoes as rising temperatures damage their physiology and overall productivity. In the 2040s, yields are expected to decline by 18–32% [3]. Cultivated potatoes are susceptible to frost and drought, reducing yield and tuber quality when the ambient temperature drops below −3 °C or the soil water potential declines to −0.3 Mpa [4,5]. Plants respond to abiotic stress through dynamic regulatory responses during transcription and protein expression, which impact various biochemical pathways and affect physiological and developmental processes at the molecular and biological levels [6]. In potato development, this results in decreased tuber yield in areas with inconsistent rainfall or poor irrigation [7,8,9]. Heat stress affects potatoes at the tuber initiation, sprout development, and photosynthesis (especially PII) activity stages; these effects limit the production and partitioning of assimilates to the sink (tuber) [10,11]. Furthermore, it impairs the nutritive quality by increasing the levels of glycoalkaloid in tubers, making the tuber bitter to consume. Responses to combinations of stresses (temperature, drought, and salinity) may differ from those to individual stresses, depending on genotypes and molecular signaling pathways [12]. Stress combinations can cause secondary stresses, such as osmotic and oxidative ones, causing cell injury and negatively impacting yield and quality [13]. Traditional breeding has failed to substantially increase potato combined-all stresses (cold, drought, heat, and salinity) tolerance, even considering the wide variation for tolerance across potato germplasm [14,15]. Plant biotechnology through genetic engineering, cisgenesis, genome edition, and omics (transcriptomic, proteomic, and metabolomic) approaches can help to improve stress tolerance. Under most stressors, it is possible to identify genes with similar regulation, even when abiotic stressors cause distinct signals and effects at the level of gene expression. Previously, we have reported a transcriptome analysis of potato microtuberization under darkness conditions, with cytokinin, and osmotic stress conditions [16,17]. Based on these findings, the STRING database v12.0 created a protein–protein interaction (PPI) network with 11 modules integrated by 404 genes. Essential cellular processes, such as ribosome biogenesis, cell cycle, and nucleosome modules, are interacting tightly with PEBP family members, carbon metabolism, oxidative stress, fatty acid, sterol biosynthesis, starch/sucrose metabolism, and secondary metabolites. We selected five INT stress-resistant genes within our PPI network and four DOMc, according to Hardigan et al., 2017 [1]. These genes include peroxidase (*PER*), enolase1 (*ENO1*), malate dehydrogenase (*MMDH*), histone *H3.2*, glyceraldehyde 3-phosphate dehydrogenase (*GAPCP1*), phosphoglycerate kinase (*PGK*), pyruvate kinase1 (*PK1*), triosephosphate isomerase (*TPI*), ribosomal protein L4 (*RPL4*), and heat shock protein 70-8 (*HSP70-8*). In contrast, other proteins such as dihydrofolate reductase *THY-1*, farnesyl pyrophosphate synthase *FPS1*, ribosomal protein L51 *RPL51*, acyl-carrier proteins I and II, S-adenosylmethionine synthase *METK2*, and superoxide dismutase/Fe *SOD/Fe* were analyzed. A two-dimensional principal component analysis (PCA) was performed to visualize the correlation between stress conditions and the relative expression of selected genes in potato microtuberization. The PCA demonstrated that 60.61% of the analyzed genes are involved in osmotic, cold–osmotic, and heat–osmotic stress. The *H3.2* gene showed the highest variance, followed by *GAPCP1.* This was followed by combined-all stresses, associated with *FPS1*, *TPI*, *RPL4*, and *SOD/Fe*, and next, salt–osmotic stress, associated with *KAS2*, *ENO1*, *HSP70-8*, and *PER*. In this manuscript, we analyzed the potential role of these genes in the solution of stress tolerance in potato microtuberization.

## 2. Results

### 2.1. Microtuber (MT) Development Under Osmotic and Combined Abiotic Stresses

The MT development of potato *S. tuberosum* cv alpha was successfully achieved in all treatments, using the MR8-G6-2iP medium (Figure 1). In the control medium (MR1-G3-2iP), no MTs developed, indicating that tuberization was reserved in the MR8-G6-2iP medium, with a high sucrose/gelrite content and 2iP, in darkness, according to Valencia-Lozano et al. (2022) [16] (Figure 1).

After fifteen days in culture, we observed differences in size and number of MTs between treatments (Figure 1, Table 1). The osmotic stress treatment yielded the biggest and most MTs in comparison to salt–osmotic, heat–osmotic, cold–osmotic, and combined-stress treatments (Table 1).

### 2.2. Identification of Stress-Responsive Genes

Two criteria were used to select stress responsive genes involved in MT development analysis: (i) the genes interacting with the essential life and developmental modules, such as nucleosome, cell cycle, and ribosomal proteins previously reported by Valencia-Lozano et al., 2022, 2023 [16,17], and Herrera-Isidrón et al., 2024 [18] (Figure 2); and (ii) a systematic review of published reports based on functional mutants and the up- and downregulation analysis focused on osmotic, heat–osmotic, cold–osmotic, salt–osmotic, and combined-all stresses analysis. Then, 17 stress-response genes were selected (Table 2). Those were genes interacting with cell cycle and carbon metabolism, such as the bifunctional dihydro-folate-reductase-thymidylate synthase-like (PGSC0003DMT400076602), S-adenosylmethionine synthase 2 (PGSC0003DMT400087679), and Heat shock 70 kDa protein 8 (PGSC0003DMT400077358); genes interacting with ribosomal proteins, such as 60S ribosomal protein L4 (PGSC0003DMT400071725M1CP75), 54S ribosomal protein L51 (PGSC0003DMT400060739), Superoxide dismutase [Fe] (PGSC0003DMT400070920), and Peroxidase (PGSC0003DMT400035521); a gene of the nucleosome module interacting with carbon metabolism: Histone H3.2-like, (PGSC0003DMT400002870); a gene of carbon metabolism interacting with nucleosomes: Glyceraldehyde-3-phosphate dehydrogenase, (PGSC0003DMT400029242); genes involved in carbon metabolism, such as Pyruvate kinase 1 (PGSC0003DMT400006945), Malate dehydrogenase, (PGSC0003DMT400032266), Enolase 1 (PGSC0003DMT400062986), Phosphoglycerate kinase (PGSC0003DMT40005687), and Triosephosphate isomerase (PGSC0003DMT400071330); genes involved in fatty acid metabolism interacting with the cell cycle, such as 3-oxoacyl-[acyl-carrier-protein] synthase II, (PGSC0003DMT400007585) and Acyl carrier protein 1 (PGSC0003DMT400036981); and a gene interacting with sterol biosynthesis: Farnesyl pyrophosphate synthase 1-like, PGSC0003DMT400076602 (Table 2, Figure 2).

Of the 17 selected genes, 9 genes were part of the list published by Hardigan et al., 2017, supplementary data S4 and S9 [1]. Of these nine selected genes, five corresponded to introgressed (INT) genes, and four to domestication (DOMc) genes. And the nine genes were identified in both landraces and cultivars (Table 2) [1]. The selected INT genes were NAD-malate dehydrogenase (PGSC0003DMG400019511), Glyceraldehyde-3-phosphate dehydrogenase (PGSC0003DMG400004130), Class III peroxidase (PGSC0003DMG400006993), Heat shock protein 70-8 (PGSC0003DMG400014835), and Pyruvate kinase (PGSC0003DMG400024220) (Table 2) [1]. The DOMc were regarded as genes under selection in both landraces and cultivars, with an impact on performance regardless of hemispheric or local adaptation. The genes were Histone H3.2 (PGSC0003DMG400001119), chloroplastic Triosephosphate isomerase (PGSC0003DMG400004436), Enolase1 (PGSC0003DMG400011044), and chloroplastic Ribosomal protein L4 (PGSC0003DMG400007051) (Table 2) [1].

Selected INT/DOMc genes published by Hardigan et al. [1] were further analyzed by reconciliation trees using Revolution-Ht software Version 2.1.2 to identify duplications and losses in gene evolution from ancestral to extant species [102] (Figure 3 and Figure 4).

The analysis encompassed seven species, including *S. commersonii* (wild type, 2x), *S. stenotomum* (landrace, 2x), *S. verrucosum* (wild type, 2x), *S. tuberosum* var. Alpha (model study), *S. tuberosum* (PGSC), *S. lycopersicum* (SoLyc), and *S. pennellii* (Sopen), which is notable for its stress tolerance. Significant evolutionary events were depicted using blue diamonds for duplications and introgressions, red circles for speciation events, and black circles without lines for losses. These markers are crucial for understanding genetic variability and species adaptation to stress.

### 2.3. Quantitative PCR Analysis of Selected Genes Under Osmotic, Heat–Osmotic, Cold–Osmotic, Salt–Osmotic, and Combined-All Stresses

A quantitative PCR analysis of stolon explants producing MTs was performed two weeks after incubation under exposure to osmotic, heat–osmotic, cold–osmotic, salt–osmotic, and combined-all-stresses environments (Figure 5). The upregulation of all analyzed genes was observed in osmotic and heat–osmotic stress treatment (Figure 5A,B). Higher levels of expression of *H3.2* were observed in osmotic stress, while *TPI*, *GAPCP1*, *RPL4*, *SOD/Fe*, and *H3.2* were found in heat–osmotic stress (Figure 5A,B). In cold–osmotic stress treatment, six genes were upregulated: *GAPCP1*, *H3.2*, *TPI*, *FPS1*, and *PK1.* The downregulation of eight genes was observed: *ENO1*, *MMDH*, *THY-1*, *METK2*, *RPL51*, *KAS2*, *ACP*, *HSP70-8*, and *PER* (Figure 5C). In salt–osmotic stress, ten genes were upregulated: *ENO1*, *PK1*, *KAS2*, *MMDH*, *HSP70-8*, *PER*, *RPL51*, *THY-1*, *PGK*, and *SOD/Fe*. The downregulation of five genes was observed (Figure 5D). In combined-all stresses, eight genes were upregulated: *GAPCP1*, *METK2*, *H3.2*, *SOD/Fe*, *RPL51*, *KAS2*, *HSP70-8*, and *PER*. The downregulation of nine genes was observed: *RPL4*, *ENO1*, *MMDH*, *PGK*, *TPI*, *PK1*, *FPS1*, and *THY-1* (Figure 5E).

### 2.4. PCA Under Different Stresses

The PCA, dimension 1 (*x*-axis, increasing average expression), demonstrates that 60.61% of the analyzed genes is involved in osmotic, cold–osmotic, and heat–osmotic stress. The *H3.2* gene exhibited the highest variance of 2.16, making it the most relevant among the stress types mentioned above, followed by the *GAPCP1*, with a variance of 2.02. This was followed by combined-all stresses, associated with the genes *FPS1* with 1.37, *TPI* with 1.00, *RPL4* with 0.78, and *SOD/Fe* with 0.48 variance; and salt–osmotic stress, associated with the genes *KAS2* with 0.21, *ENO1* with 0.19, *HSP70-8* with 0.16, and *PER* with 0.13 variance (Figure 6A).

In dimension 2 (*y*-axis, increasing positive trend), the PCA explains that 21.14% of the variance tends to a positive expression of the genes. They have a direct relationship to the abiotic stress involved in MT development. These genes correspond to *H3.2* and *GAPC1* (Figure 6A). Corr PCA shows that while dimension 1 (PC1) captures the overall variability in gene expression across all treatments, dimension 2 (PC2) emphasizes the specific contributions of genes related to osmotic stress adaptation. The higher importance of these genes in PC2 suggests a clear differentiation in the adaptive response to osmotic stress compared to other types of stress represented in the feature space, underscoring their crucial role in adaptation to water scarcity (Figure 6B).

In this study, the variance was used in each component to comprehend the network of correlations between various stresses during the MT development of potatoes under darkness.

As the variance increases in the treatment, its value increases. Osmotic, heat–osmotic, cold–osmotic, and combined-all stresses have a major impact on gene regulation. Salt–osmotic stress does not interact with the treatments (Figure 7).

The gene set enrichment analysis was used to clarify the overall activity under various stresses. Notably, genes associated with osmotic, cold–osmotic, and heat–osmotic stress (*H3.2* and *GAPCP1*) have a key role in abiotic stress adaptation, exhibiting distinctive regulatory dynamics and potential functional implications across different stresses (Figure 8).

Osmotic stress interacts mainly with heat–osmotic and cold–osmotic stresses; heat–osmotic with cold–osmotic and combined-all stresses, and cold–osmotic with combined-all stresses, while salt–osmotic keeps a low interaction. Osmotic, cold–osmotic, and heat–osmotic stress significantly impacts gene expression, regulating key genes (*H3.2* and *GAPCP1*) involved in adaptation to abiotic stress, followed by combined-all stresses (*FPS1*, *TPI*, *RPL4*, and *SOD/Fe*). Additionally, the salt–osmotic treatment interacts with all variables, but the activated genes (*KAS2*, *ENO1*, *HSP70-8*, and *PER*) do not play a direct role in resistance to other types of stress. Instead, they are important for a specific adaptation to salt–osmotic stress (Figure 8).

### 2.5. Cis-Acting Elements Present in the Genes with the Highest Variance in Different Stresses

*H3.2* was mostly expressed under heat–osmotic stress (6.81), followed by osmotic stress (5.93) and cold–osmotic stress (1.64). *H3.2* does not have described motifs that confer heat (STRE and CCAAT-box) or cold (LTR) stress. However, osmotic stress-related motifs like MYB at position 308 have a slight effect on heat and cold stress [18] (Figure 9). *GAPCP1* was expressed under heat (8.16), followed by osmotic stress (4.28). The STRE motif is in the middle of the gene at position 812, while the osmotic stress motif (MYB) is at position 312 and MBS at position 53 (Figure 9). Heat (6.87), osmotic (4.43), and cold stress (1.63) were the conditions under which *FPS1* expressed itself most. The STRE motif is located at position 280, while MYB is at position 390 (Figure 9). *TPI* showed increased expression under heat (8.4), osmotic (2.37), and cold stress (1.63). The STRE gene position is near the origin at 212, and MBS at position 27 (Figure 9). *RPL4* showed increased expression under heat (7.69) and osmotic stress (3.1). The STRE motif, located at position 60 at the beginning of the gene, and MYB motif, at 430, exhibited a strong correlation (Figure 9). *SOD/Fe* showed increased expression under heat (7.51), combined-all stresses (4.12), and cold stress (0.87). The STRE motif is located at position 430, conferring resistance to heat stress. The CCAAT-box at 498 and the TC-rich repeats at position 571 are involved in stress response. MYC at 302 and MYB at 836 are involved in osmotic/salinity stress. The LTR motif is located at position 867 for cold stress (Figure 9).

*KAS2* showed increased expression under salt–osmotic (5.65), heat (2.45), and osmotic stress (1.24). The MYC motif is at position 65, MYB at 431, and for heat stress, the TC-rich repeats are at position 992 (Figure 9). *ENO1* showed increased expression under osmotic (3.25), followed by heat stress (2.28). The MYB motif in *ENO1* is at position 35, which is closely linked to its function under osmotic stress. The STRE motif is at position 383 (Figure 9). *HSP70-8* showed increased expression under salt–osmotic (4.89) and osmotic stress (0.57). It has 11 MYB motifs in the sequence starting at position 838 (Figure 9). *PER* showed increased expression under salinity (4.46), heat (4.25), and osmotic stress (2.75). The motifs related to salinity/osmotic stress, two MYB and four MYC, are located at positions 607 and 631 near the origin of replication; the STRE motif is at position 273, and the LTR (cold) motif is at position 1102 (Figure 9).

## 3. Discussion

Global consumption ranks potato (*S. tuberosum* L.) as the fourth most significant food crop [103]. It is a temperate staple food crop that produces modified stems called tubers, rich in energy and proteins. This crop was domesticated 8000–10,000 years ago from its wild ancestor species in the Andes of southern Peru [1]. Early domestication involved choosing tubers that were not non-bitter (high content of glycoalkaloids) [104]. These tubers were then adapted to different Andean environments, ranging from frost-tolerant species that could growth in dry, high altitudes areas to varieties that could grow in lower mountain valleys. Climate change, including rising temperatures, decreased precipitation, and salinity, negatively impacts plant growth, development, and productivity. A temperature rise of 1.6–3 °C could lead to a decrease in worldwide potato production by 18–32% [3].

Plants regulate the coordinated expression of numerous stress-related genes in response to environmental stress. Understanding the interaction between molecular, physiological, and biochemical mechanisms for stress adaptation is crucial for developing potato management techniques that aid in adaptation to abiotic stress in the context of climate change.

In this work, we analyzed seventeen genes derived from a PPI network with high confidence (0.800), based on a transcriptome analysis of potato microtuberization under darkness [16]. Of these, five were identified as INT and four as DOMc during potato domestication from wild ancestors [1]. Interestingly, the PCA revealed that seven out of nine INT and DOMc genes showed the highest variance in osmotic stress, heat, and cold stress (Figure 6). Accordingly, *H3.2*, with a variance of 2.16, was the most important of the above stress types, followed by *GAPCP1* with a 2.02 variance. The next most significant genes in response to combined-all stresses were *FPS1* with 1.37, *TPI* with 0.93, *RPL4* with 0.78, and *SOD/Fe* with 0.48 variance. In response to salt–osmotic stress, the most significant genes were *KAS2* with 0.21, *ENO1* with 0.19, *HSP70-8* with 0.16, and *PER* with 0.13 variance.

In the reconciliation trees, a correlation between more duplications and fewer losses can be observed with the relevance of the PCA. *H3.2* (Figure 4) was the most important gene in the PCA and showed many duplication genes and few losses in the reconciliation tree. Conversely, *PK1* and *MMDH* (Figure 3), which were not relevant in our PCA, displayed many losses and few duplication genes.

Osmotic/heat–osmotic/cold–osmotic resistance genes with higher variance/importance were involved in MT development.

### 3.1. Role of Histone H3.2, Nucleosome, DNA Priming, and Memory Stress

In our PPI network derived from the transcriptome analysis of potato microtuberization, the nucleosome module is composed of 11 genes. This module interacts tightly with the cell cycle, ribosome biogenesis, and carbon metabolism (Figure 2).

The nucleosome consists of proteins that package and protect the DNA of all organisms. It is composed of four core histones, which are universal, and the architecture is invariant across eukaryotes [19]. Repeating units of nucleosomes are thread-like stained bodies called chromatin. Plants have challenged the adverse environment through chromatin remodeling, thus facilitating plant growth and development. Osmotic stress has been related to chromatin remodeling [20,21], salinity [22], and extreme temperatures [23].

Histones and epigenetic mechanisms play crucial roles in adaptation to various environmental stressors. Histones are key components of chromatin and regulate gene expression through epigenetic modifications such as methylation and acetylation.

Tri-methylation on H3K4 (H3K4me3) and the modification of histone H3.2 are important for responses to stress. In potato tubers, the meristematic activity is activated by the increased acetylation of histones *H3.1* and *H3.2* and transient increases in *H4* multi-acetylation [24]. Potato cold stress enhanced the chromatin accessibility, and histone modifications H3K4me3 and H3K27me3 were enhanced [25]. H3K27 methyltransferase regulates the expression of key tuberization genes such as *StBEL5/11/29*, *StSWEET11B*, *StGA2OX1*, *StSP6a*, and *StPIN1* in potatoes. The overexpression line construct showed a reduction in tuber yield, while its knockdown increased yield, suggesting a role in the tuberization process [26]. Rice roots under salt–osmotic stress showed an upregulation of the RH3.2A gene, and seedlings treated with ABA showed the same response [27].

Overexpressing histone *H3.2* in tomato showed a reduction in cell number and increased cell size, leading to the growth retardation, a similar phenotype to *CYCLIN-B1*, both of which regulate cell cycle and subsequently plant growth [28]. In rice, cold stress is attenuated by *H3.2* [29] and osmotic stress [30].

### 3.2. Generation of Energy and Primary Metabolites Through GAPCP1

GAPCP plays a specific role in glycolytic energy production in nongreen plastids and is absent in the chloroplasts of angiosperms [31]. This enzyme is essential for starch metabolism during the dark period in both green and nongreen plastids. GAPCP, along with the phosphoglycerate kinase, has been demonstrated to be involved in the production of ATP needed for both starch metabolism and biomolecule synthesis pathways [32]. In our protocol, microtuberization was activated in darkness, indicating the involvement of GAPCP and phosphoglycerate kinase in the starch metabolism of tubers.

The overexpression of *GAPCP1* enhances osmotic stress tolerance in potato [33] and salt tolerance in potato (*S. tuberosum*) [34], soybean (*Glycine max*) [35], and rice (*Oryza sativa*) [36]. In *Arabidopsis thaliana*, double mutants exhibit a drastic phenotype of arrested root development, dwarfism, and sterility [37]. *TaGAPCp1* plays an important role in wheat’s response to osmotic stress via the ABA signaling pathway. TaGAPCp1 interacts with Cytb6f. It was speculated that the stress resistance process of *TaGAPCp1* might probably be completed by the H_2_O_2_-mediated ABA signaling pathway with H_2_O_2_ acting as a signal molecule, while the antioxidant activity of carotenoids in *Cytb6f* could probably maintain the relative balance of ROS. These findings demonstrate that *TaGAPCP1* is a critical factor in the abiotic stress responses in wheat [38]. The overexpression of *GAPCP1* in *A. thaliana* exhibited improved morphological parameters and the accumulation of photosynthetic pigments compared to wild-type (WT) plants under salinity stress conditions [39]. The co-suppression of these three GAPC genes resulted in low tuber GAPDH activity and, consequently, the accumulation of reducing sugars in cold-stored tubers by altering the tuber metabolite pool sizes, favoring the sucrose pathway [40].

The overexpression of *StGAPC1* promoted potato seedling growth and nitrogen accumulation under N starvation stress [41]. The overexpression of this gene promotes heat tolerance in *A. thaliana* [42] and rice [43].

### 3.3. Combined-All Stresses

#### Farnesyl Pyrophosphate Synthase FPS1 Isoprenoids Biosynthesis

The downregulation of the *FPS* gene in *A. thaliana* by the miRNA of the *FPS* gene results in a chlorotic phenotype, an altered profile of cytosolic and plastidial isoprenoids, and sterol depletion. Plants perceive this sterol depletion as a stress signal, which triggers early transcriptional stress responses, such as Jasmonic acid (JA) signaling and Fe homeostasis. FPP functions as a precursor of several essential isoprenoid end products, with sterol depletion being the primary cause of the observed alterations [18,44].

In this work, several genes involved in abiotic stress were downregulated: freezing tolerance, Low-temperature-responsive protein 78 (*LTI78*), Cold-regulated 15b, Cold-regulated 413 inner membrane protein 1, Chilling tolerance, and fructose-1, 6-bisphosphate aldolase [45]. To enhance tuber quality, *CDGSH* (2Fe–2S) containing the protein NEET has been used [46]. MLPs play crucial roles in numerous abiotic stresses containing osmotic and salt stress and resistance against pathogens, including infectious fungi, bacteria, viruses, and phytoplasma, by the induction of defense-related genes [47].

In potato, the overexpression of *StHMGR1*, *StHMGR3*, *StHMGR1/StFPS1*, and *StHMGR3/StFPS1* resulted in changes in the expression of sterol biosynthesis genes, which affected flowering, stem height, biomass, and tuber weight [48]. It also regulates the transitions of flowering and tuberization in potato [49]. Double mutant *fps1/fps2* is embryo-lethal [50]. The overexpression of *FPS* confers heat and cold in *Solanum viarum* [51], osmotic stress and heat tolerance in tobacco [52], and salt stress in *Rosa rugosa* [53].

It confers resistance to heat and cold stress in *S. viarum* [51], heat and osmotic stress in *Dryopteris fragrans* [54], and cold tolerance in apple [55].

### 3.4. Triosephosphate Isomerase, TPI: The Perfect Catalyst

TPI catalyzes the interconversion of the glycolytic intermediates dihydroxyacetone phosphate (DHAP) and glyceraldehyde-3-phosphate (GAP). TPI activity is ubiquitous in both prokaryotes and eukaryotes. TPI is generally considered an extremely efficient enzyme and referred to as a perfect catalyst [56]. TPI enhances photosynthesis under elevated CO_2_ levels in rice, makes pigeon peas more resilient to salt stress [57], and improves osmotic stress tolerance in rice [58] and maize [59].

Single mutants *tpi1*, *tpi2* had no visible phenotypes, whereas double mutants *tpi1*/*tpi2* had reduced *TPI* activity and displayed chlorotic variegation, and lower carbon-assimilation efficiency, severely affecting photosystem proteins, reducing photosynthetic capacity [60].

However, the two proteins differed in their responses to heat stress. The protein encoded by the heat-induced *SlTPI2* showed a higher level of thermotolerance than that encoded by the heat-suppressed *SlTPI1* [60]. In potatoes, high salinity greatly decreased the expression of genes involved in photosynthesis and primary metabolism, which includes carbohydrate-related genes like Glyceraldehyde-3-phosphate dehydrogenase (*G3PDHase*) and *TPI* in potato. Both enzymes catalyze important reactions in the glycolytic pathway, which make building blocks for biosynthetic processes that use a lot of energy [61].

### 3.5. Ribosomal Protein 4, RPL4

The *RPL4* mutants display an abnormal transition from the globular to the heart stage of embryogenesis [62]. Yeast transformants expressing potato *RPL4* cDNAs demonstrated resistance to osmotic, salt, and heat stress [63]. The *RPL4* gene was upregulated in heat and cold stress in rice [64]. *RPL4* is involved in osmotic stress tolerance in spinach and *A. thaliana* [65]. The *rpl4* mutants display a range of growth abnormalities, altered cotyledon architecture, vacuolar sorting defects, and antibiotic resistance, and the mutant *rpl4a/rpl4d* display embryonic lethality [66]. The knockout of *RPL4* in *Nicotiana benthamiana* causes chlorosis and stunted growth [67]. Furthermore, rice responds to heat and cold stress by upregulating the *RPL4* gene [64].

### 3.6. Superoxide Dismutase, FeSOD, Is the First SOD to Evolve Due to the Abundance of Iron and Low Levels of Oxygen in Earth’s Primitive Atmosphere

Plants respond to various stresses to scrub the reactive oxygen species by producing enzymatic and nonenzymatic molecules, including catalases, peroxidases, and superoxide dismutases [68]. The activities of SOD and peroxidase were higher in potato with salt- treated potatoes than untreated potatoes [69], and improved cold tolerance in potato [70], cassava [71], and rapeseed [72]. SOD/Fe levels in *Nicotiana plumbaginifolia* plants remain stable under prolonged darkness, whereas under light exposure, there is a significant increase in SOD/Fe production [73]. SOD enzymes are expressed highly in rice anther [74] and wheat under heat stress [75].

### 3.7. Salt–Osmotic Stress

#### 3-Oxoacyl-[Acyl-Carrier-Protein] Synthase II, KAS2

KAS2 is an essential protein that catalyzes the condensation reaction of fatty acid synthesis by the addition of an acyl acceptor of two carbons from malonyl-ACP. It is specifically designed to elongate C-16 fatty acid into unsaturated C-18 fatty acids. It confers resistance to low temperatures by maintaining chloroplast membrane integrity. It is involved in the regulation of fatty acid ratios during seed metabolism. It is required for embryonic development, especially at the transition from the globular to the heart stage. Halophyte algae, *Dunaliella salina* [76], and *Chromochloris zofingiensis* [77] have shown a high expression of this protein.

### 3.8. Enolase 1, ENO1

Enolases, also known as 2-phospho-D-glycerate hydrolases, are highly significant enzymes in the glycolysis process. These enzymes have traditionally been recognized for their ability to facilitate the removal of water from 2-phosphoglycerate, resulting in the formation of phosphoenolpyruvate. The enolase mutant is sensitive to salt and osmotic stress in *A. thaliana* [78]. ENO1 binds with the heat shock protein *HSP70* and confers heat tolerance [79]. The homozygous *los2* mutant exhibited significant growth impairment, pale green color, and failed flowers, siliques, and seeds [80].

### 3.9. Heat Shock 70 kDa Protein 8, HSP70-8

In cooperation with other chaperones, HSP70s are key components that facilitate folding of de novo synthesized proteins, assist the translocation of precursor proteins into organelles, and are responsible for the degradation of damaged proteins under stress conditions.

In potato, *StHSP70-1*, *StHSP70-8*, *StHSP70-9*, *StHSP70-10*, and *StHSP70-17* were upregulated in osmotic, heat, cold, and salt stresses, and *StHSP70-8* was the only one found to be highly expressed under the hormone treatment assays (ABA, IAA, GA_3_, and SA) [81].

*NtHSP70-8* confers osmotic stress tolerance in tobacco by regulating water loss and antioxidant capacity, as well as the interaction between ABA and auxin signaling [82]. *NtHSP70-8* is highly expressed under heat stress, and its overexpression increased the seed size of tobacco, thus affecting the 1000-seed weight [54]. In wheat durum, salt-stressed landraces demonstrated decreased growth, increased levels of stress indicator parameters, and upregulation in *HSP17.8*, *HSP26.3*, *HSP70*, and *HSP101* expressio*n* [83].

*HSP70* plays a critical role in the cellular homeostasis of plants during adaptation to osmotic stress. *HSP70* silencing led to severe growth retardation and mortality, significant membrane damage and leakage, a decline in relative water content, a low rate of pigment accumulation, and reduced antioxidant enzyme activity under normal and osmotic stress conditions [84].

### 3.10. Peroxidase, PER

Salt-stressed potato plants showed an increase in glutathione metabolism, peroxisome, flavonoid biosynthesis, and ascorbate oxidase. *CAT3*, *SOD*, *PER7*, and *PER66* were all upregulated under salt stress treatment [85]. Peroxidase is highly induced in salt and oxidative stress in sweet potato [86], tomato [87], cotton [88], bean [89], and cucumber [90]. Also, it is induced in heat stress in strawberry [91] and apple [92], and in osmotic stress in wheat [93], walnut trees [94], and barley varieties [95].

Gamma-treated potatoes exhibited tolerance to salinity by increasing the levels of peroxidase and polyphenol oxidase [96]. The activities of SOD, peroxidase, and ascorbate peroxidase were higher in salt-treated potatoes [69].

We proposed a model from the analyzed data set, which included the DEG and qPCR transcriptional expression levels of DOMc, INT, and stress resistance genes in different conditions. The PCA revealed that the ancestral genes associated with DOMc and INT were more important than the rest of the analyzed genes (Figure 10).

In osmotic, heat, and cold stress, *H3.2* (DOMc) was the most important with 2.15 variance, followed by *GAPCP1* (INT) with 1.93 variance. In terms of combined-all stresses, *FPS1* was the most important, with 2.26 variance, followed by *TPI* (DOMc) with 1.64 variance, then *RPL4* (DOMc) with 1.28 variance, and *SOD/Fe* with 0.79 variance.

In salt stress, *KAS2* was the most important with 0.21 variance, followed by *ENO1* with 0.18 variance, then *HSP70-8* with 0.16 variance, and finally *PER* with 0.13 variance (Figure 10). In accordance with tuber development under natural conditions, MT size and number change according to the stress type. In osmotic stress, MTs are bigger and produce a higher number compared to salt–osmotic, heat–osmotic, cold–osmotic, and combined-all stresses (Figure 10).

Further experiments will help to elucidate the practical use of INT, DOMc, and the other stress resistance genes analyzed through the application of cisgenesis, and genome edition will help us to elucidate it.

## 4. Materials and Methods

### 4.1. Plant Material and MT Induction

Potato *S. tuberosum* cv. Alpha was used in this work. MT induction was carried out according to [105]. Stolon with two internodes were cultured in induction medium MR8-G6-2iP (8% *w*/*v* sucrose, 6 g/L gelrite, and 10 mg/L 2iP). The control medium utilized was MR1-G3-2iP (1% *w*/*v* sucrose, 3 g/L gelrite, and 10 mg/L 2iP). Containers were stored in the dark at 25 °C/17 °C for 15 days.

To induce salt stress, NaCl 50 mM was added to the medium, MR8-G6-2iP (osmotic stress combination). For temperature stresses, the first 24 h containers containing stolon explants were incubated at 4 °C and 38 °C for cold–osmotic and heat–osmotic stress, respectively. For combined-all stresses, explants cultured in MR8-G6-2iP (osmotic stress) plus NaCl 50 mM (salt stress) were subjected to heat (38 °C) for 24 h, followed by cold (4 °C) for 24 h, and the rest for 13 days at 25 °C/17 °C (Figure 11).

### 4.2. Isolation of RNA, qPCR, and Transcriptome Sequencing

Total RNA isolation was performed using Trizol reagent (Invitrogen, Carlsbad, CA, USA). The RNA concentration was determined by measuring its absorbance at 260 nm, and the ratio (260 nm/280 nm) of absorbance was assessed. The RNA integrity was confirmed by agarose gel electrophoresis at 2% (*w*/*v*). The cDNA samples were amplified using SYBR™ Green (ThermoFisher CAT: 4312704, Waltham, MA, USA) in Real-Time PCR Systems (CFX96 BioRad, Hercules, CA, USA).

### 4.3. Analysis of DEG and Interaction Analysis of Stress Genes

A gene network was constructed using STRING [106] with a confidence score of 0.800. The network was built including homologous genes found in the *S. tuberosum* genome from the Sol Genomics Network. A gene identifier (Id) was based on the UNIPROT [107] and NCBI databases [108]. Protein sequences in *S. tuberosum* that have an over 60% similarity with *A. thaliana* were considered.

### 4.4. Reconciliation Trees

Validation was achieved using Revolution-Ht software to reconstruct reconciliation trees, a method that infers the evolutionary history of a gene family by overlaying gene trees onto a known species tree. This approach facilitates the identification of duplications and losses in gene evolution from ancestral to extant species [102].

### 4.5. Transcriptional Analysis Through qPCR of Genes Involved in Stress Response

Table 3 presents the list of INT/DOMc and others stress-resistant genes that were analyzed using qPCR. The genes *EF1* and *SEC3* were used as benchmarks to determine the relative expression of the genes of interest, applying the 2^−∆∆CT^ technique [109]. Each sample was subjected to analysis with five biological replicates, and each was tested three times technically during the qPCR process.

### 4.6. Principal Component Analysis of Gene Expression

The PCA method was used for the data of gene expression under five different stress conditions. The relative expression was subjected to PCA after standardizing to unit variance. The resulting factor scores of PC1 and PC2 were tested in a two-way analysis of variance (ANOVA). Data analyses were carried out using Rstudio [110].

### 4.7. Analysis of Cis-Acting Regulatory Elements in Genes

The Plant CARE database [111] was utilized to identify potential cis-acting regulatory elements within the promoter regions spanning 1 to 2000 base pairs upstream of the transcription start sites of the genes under study and visualized in a heat map using the TBtools software [112].

## 5. Conclusions

The PCA demonstrated that 60.61% of the variability in gene expression is associated with responses to osmotic, cold, and heat stress. The genes with the highest variance were *H3.2* (DOMc) and *GAPCP1* (INT). Chromatin remodeling and histone modification interacting with the carbon metabolism are involved in MT development under conditions of osmotic, heat, and cold stress.

Salt–osmotic stress response genes were related to *KAS2*, *ENO1* (DOMc), *HSP70-8* (INT), and *PER* (INT). Peroxidase protects the cytoplasm from ROS induced by salt stress; *KAS2* links carbon metabolism and fatty acid biosynthesis, thereby protecting membranes during salt stress; and HSP70-8 controls antioxidant enzymes and membrane integrity.

The analysis of cis-acting elements revealed that the gene *H3.2* contains motifs related to osmotic stress (MYB) and heat, although not specifically to cold. *GAPCP1* and *FPS1* showed high expression under heat and osmotic stress, with STRE and MYB motifs present. *SOD/Fe* exhibited high expression under heat–osmotic and combined-all stresses, with STRE and LTR motifs associated with resistance to multiple stress types.

Four INT genes and three DOMc genes were identified as pivotal contributors associated with the potato response to environmental stress and as crucial contributors to enhancing stress tolerance. These genes exhibit significant variability and importance in managing osmotic, heat–osmotic, cold–osmotic, salt–osmotic, and combined-all stresses.

## Figures and Tables

**Figure 1 plants-13-02996-f001:**
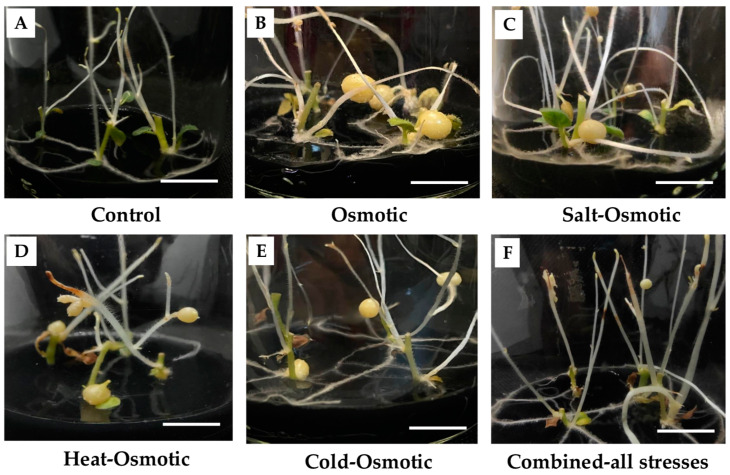
(**A**): Stolon explants of potato in control medium MR1-G3-2iP; no MTs were observed. (**B**–**F**): MT development in potato *S. tuberosum* cv. Alpha, after fifteen days in culture in MR8-G6-2iP medium (osmotic stress) plus NaCl 50 mM (salt–osmotic stress), MR8-G6-2iP exposed at 38 °C for 24 h (heat–osmotic stress), MR8-G6-2iP exposed at 4 °C for 24 h (cold–osmotic stress), and the combination of osmotic stress in MR8-G6-2iP exposed to NaCl 50 mM, followed by heat–osmotic and cold–osmotic stress. Scale bar represents 1 cm.

**Figure 2 plants-13-02996-f002:**
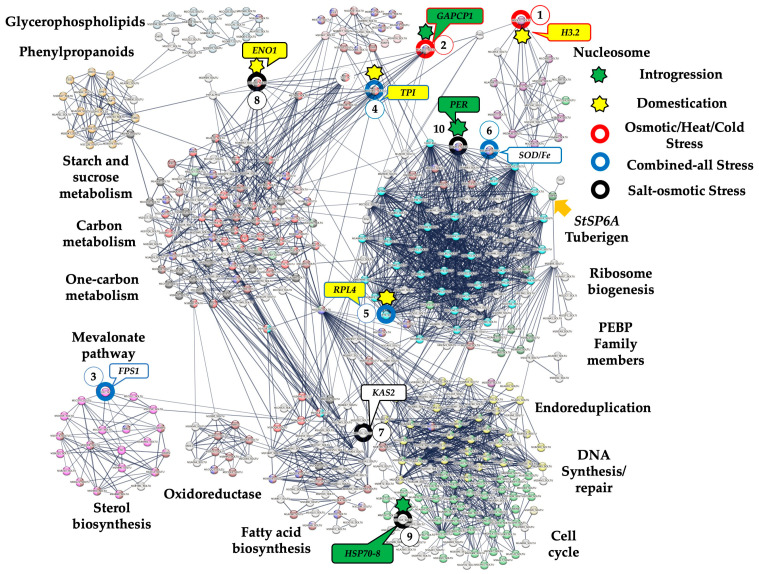
PPI network of upregulated genes derived from the STRING database v12.0 of potato *S. tuberosum* from the transcriptomic-wide analysis with high confidence (0.800). Circles are related to the most important genes in different stresses: red for osmotic/heat/cold, blue for combined-all stresses, and black for salinity. Numbered circles correspond to the level of importance according to the PCA in different stresses. Green stars represent INT genes from wild-type ancestors during potato domestication. Blue stars represent genes involved in DOMc from landraces to cultivated potatoes.

**Figure 3 plants-13-02996-f003:**
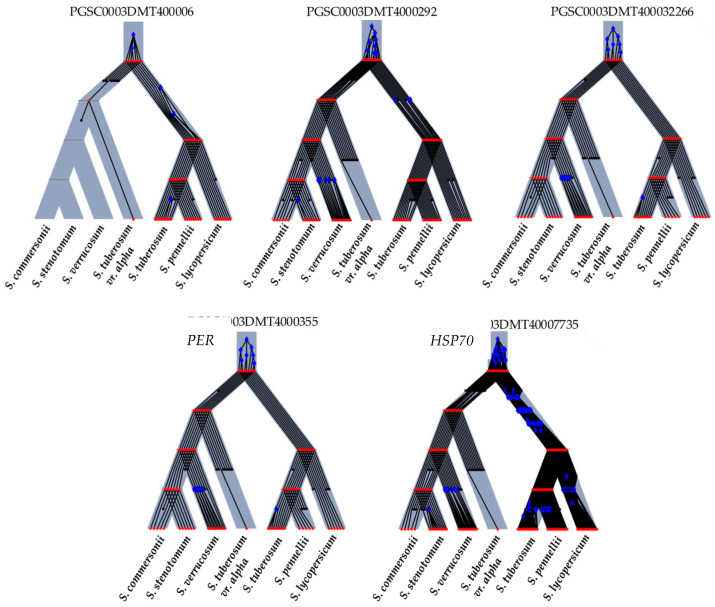
Reconciliation trees of introgressed genes. As can be seen in all trees, the INT genes were conserved in the evolution of species. PGSC0003DMT400006945 (M0ZS78—*PK1*—Pyruvate kinase 1), PGSC0003DMT400029242 (M1ASG7—*GAPCP1*—Glyceraldehyde-3-phosphate dehydrogenase), PGSC0003DMT400032266 (M1AX44—*MMDH*—Malate dehydrogenase), PGSC0003DMT400035521 (M1B2E4—*PER*—Peroxidase 7), PGSC0003DMT400077358 (M1CYA5—*HSP70-8*—Heat shock 70 kDa protein 8). Blue diamonds (duplications or introgressions). Red circles (speciations). Black circles without lines (losses).

**Figure 4 plants-13-02996-f004:**
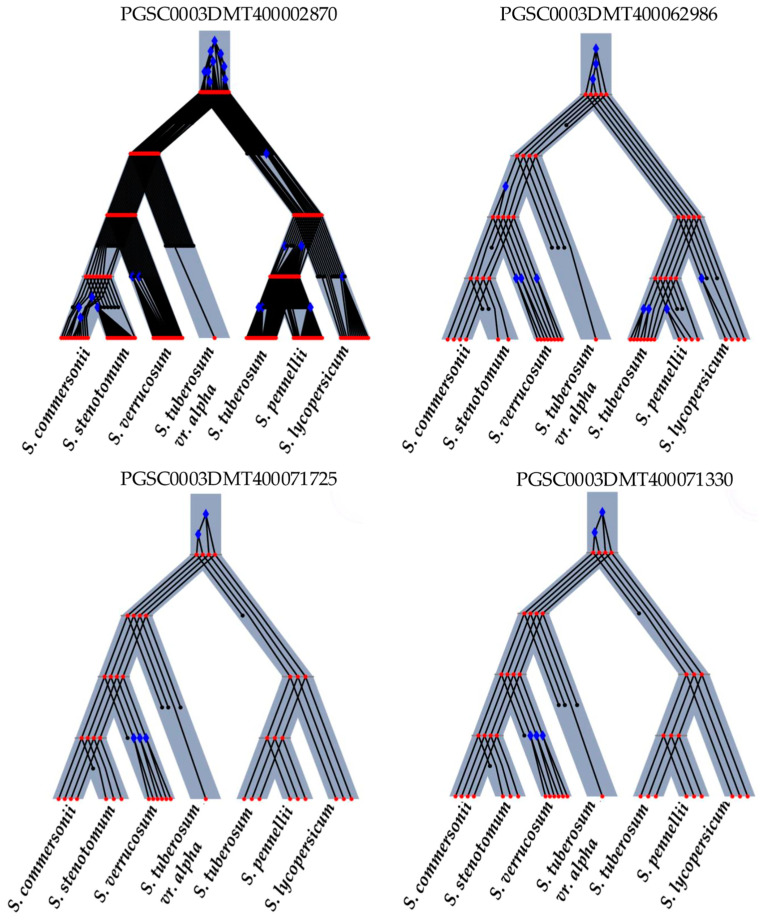
Reconciliation trees of domesticated genes: PGSC0003DMT400002870 (M0ZKT2—*H3.2*—Histone H3.2-like), PGSC0003DMT400062986 (M1C9X0—*ENO1*—Enolase 1), PGSC0003DMT400071725 (M1CP75—*RPL4*—60S ribosomal protein L4-1-like), PGSC0003DMT400071330 (M1CNK1—*TPI*—Triosephosphate isomerase). Blue diamonds (duplications or introgressions). Red circles (speciations). Black circles without lines (losses).

**Figure 5 plants-13-02996-f005:**
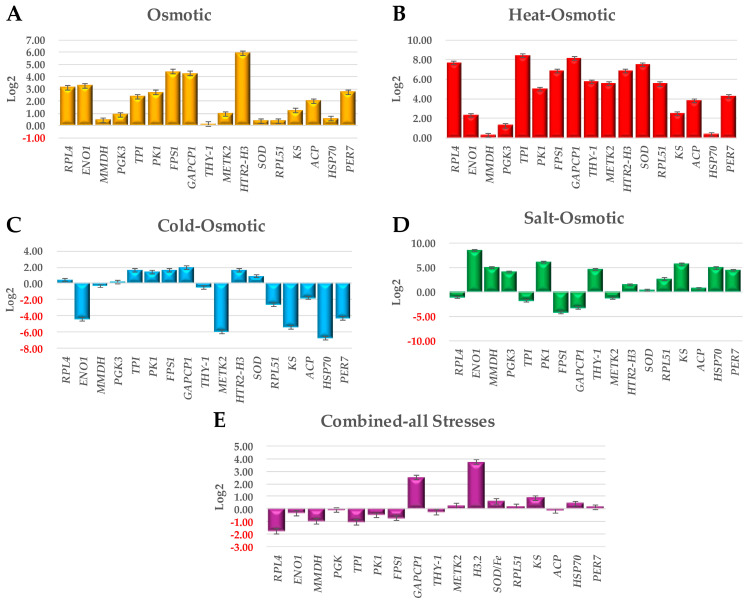
Quantitative PCR analysis of the stolon explants producing MTs of seventeen selected genes under osmotic (**A**), heat–osmotic (**B**), cold–osmotic (**C**), salt–osmotic (**D**), and combined-all stresses (**E**). Relative expression estimation levels are represented in Log2-Fold Change.

**Figure 6 plants-13-02996-f006:**
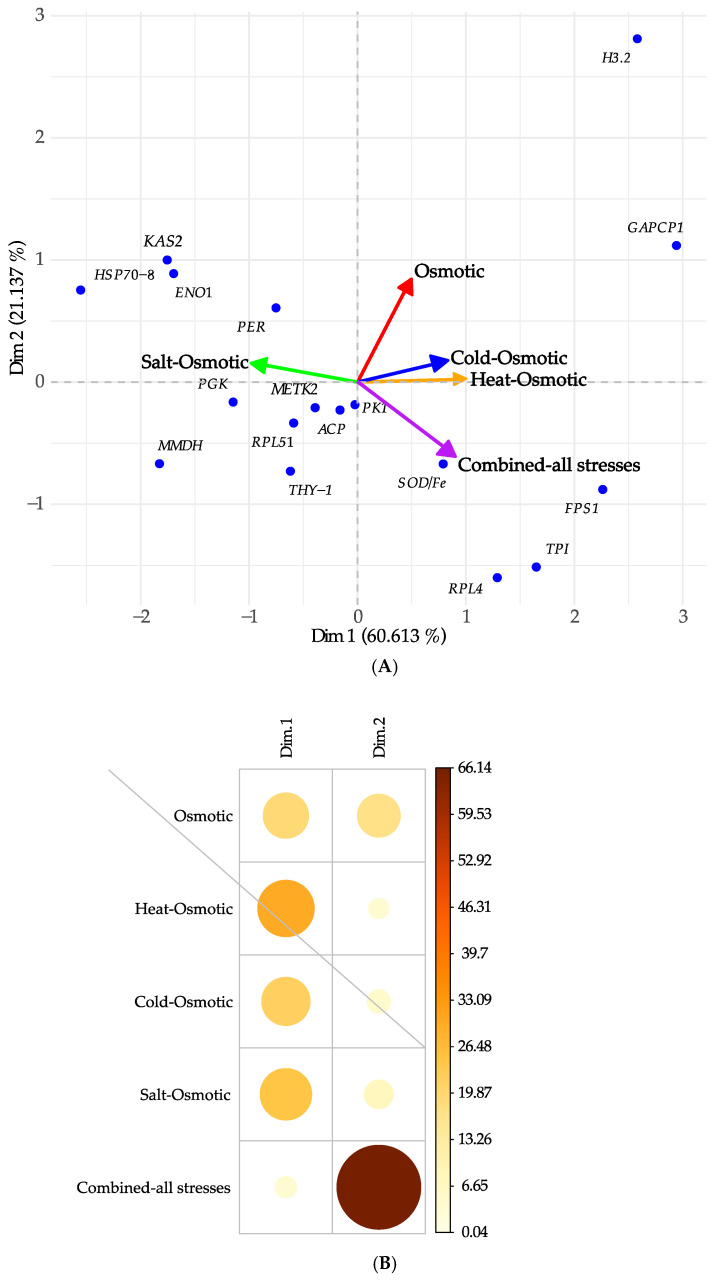
(**A**) Principal component analysis (PCA) from the relative gene expression under different type of stresses during the microtuberization of potato under darkness. The group of genes with the highest variance were *H3.2* and *GAPCP1*, involved in osmotic, cold–osmotic, and heat–osmotic stress. Combined-all stresses was associated with *FPS1*, *TPI*, *RPL4*, and *SOD/Fe*, and salt–osmotic stress was associated with *KAS2*, *ENO1*, *HSP70-8*, and *PER*. (**B**) In Corr PCA, dimension 1 (PC1) shows a uniform distribution in the number of genes, indicating a general variability in gene expression without bias toward a specific treatment. In contrast, dimension 2 (PC2) reveals a high load of genes associated with osmotic stress response, highlighting their predominant relevance in this dimension.

**Figure 7 plants-13-02996-f007:**
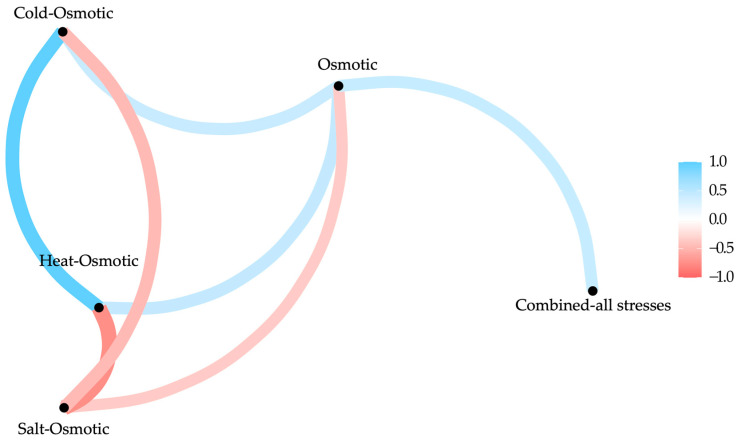
Correlation network between different stresses during the microtuberization of potato under darkness. Variance was used in each component, as the greater the variance in the treatment, the greater its value. Heat–osmotic, cold–osmotic, osmotic, and combined-all stresses have a major impact in gene regulation. Salt–osmotic stress is not interacting with the others.

**Figure 8 plants-13-02996-f008:**
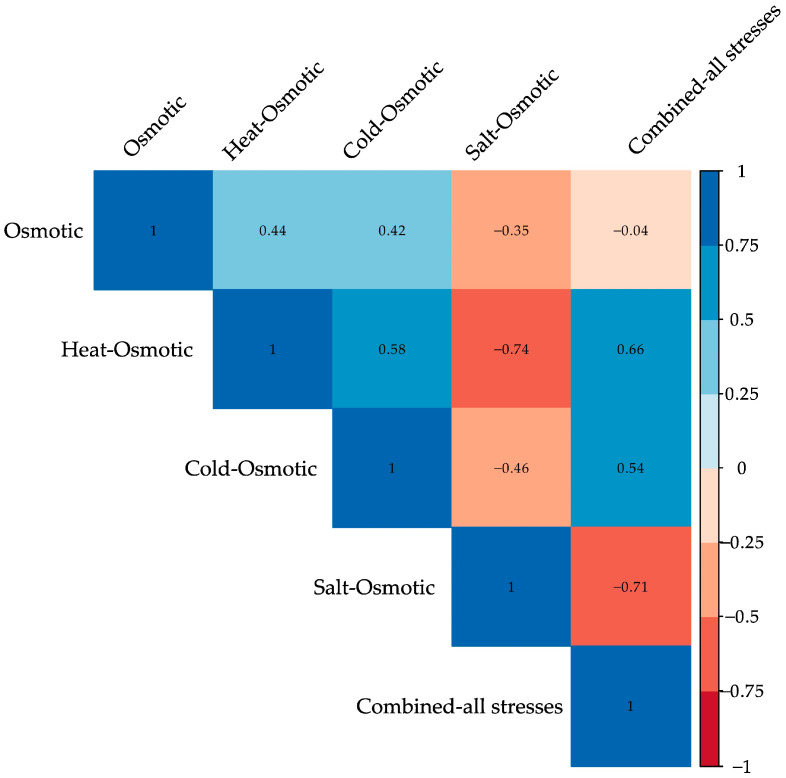
Heat map of the variable contribution of genes during MT development in potato exposed to different stresses. The gene set enrichment analysis elucidates the overall activity under different stresses, like osmotic, heat–osmotic, cold–osmotic, salt–osmotic, and combined-all stresses. Notably, genes associated with osmotic, cold–osmotic, and heat–osmotic stress (*H3.2* and *GAPCP1*) have a key role in abiotic stress adaptation, exhibiting distinctive regulatory dynamics and potential functional implications across different stresses.

**Figure 9 plants-13-02996-f009:**
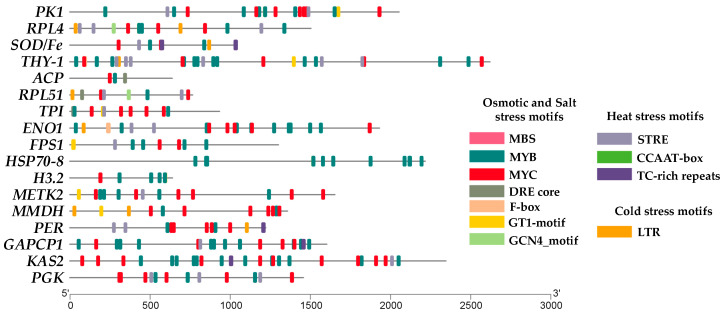
Cis-acting elements present in the genes with the highest variance in different stresses during the MT development of potato.

**Figure 10 plants-13-02996-f010:**
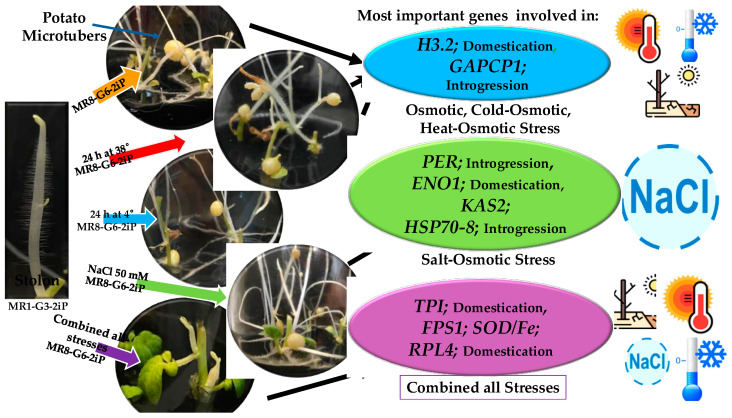
A model proposal for the analysis of stress response genes in the microtuberization of potato *S. tuberosum*. Contributions to osmotic, heat–osmotic, cold–osmotic, salt–osmotic, and combined-all stresses tolerance. Osmotic/heat–osmotic/cold–osmotic stress (blue), salt–osmotic stress (green), and combined-all stresses (violet).

**Figure 11 plants-13-02996-f011:**
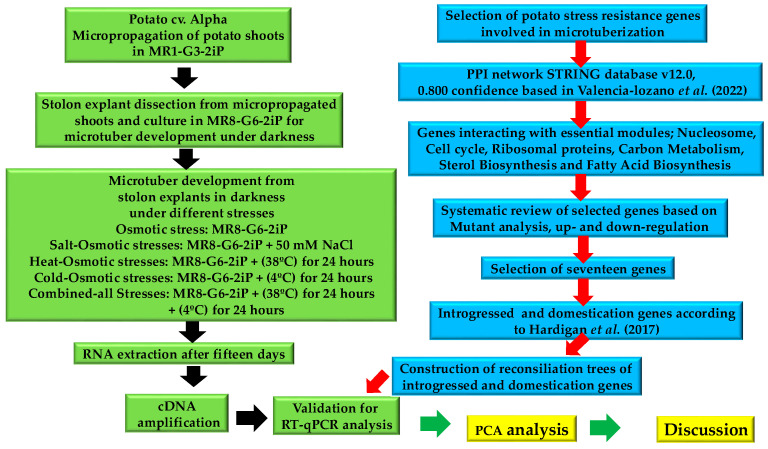
Methodology of microtuberization of potato *S. tuberosum* cv. Alpha under different stresses and genetic analysis of stress resistance genes.

**Table 1 plants-13-02996-t001:** Analysis of diameter and number of MTs produced in osmotic and combined abiotic stress treatments after 15 days in culture.

Stress	MTs Diameter (mm)	MTs Per Explant Mean ± SE
Osmotic	3.60 ± 0.13	0.85 ± 0.11
Heat–Osmotic	3.04 ± 0.08	0.75 ± 0.24
Cold–Osmotic	2.90 ± 0.07	0.70 ± 0.21
Salt–Osmotic	3.57 ± 0.09	0.75 ± 0.35
Combined-all stresses	2.62 ± 0.06	0.35 ± 0.14

**Table 2 plants-13-02996-t002:** Upregulated genes selected for analysis in osmotic, heat, cold, salinity, and combined-all stresses.

Potato ID String v11.5	Potato ID String v12.0	*A. thaliana*	INT/ DOMc	Annotation	Reference Systematic Review
*PGSC0003DMT400006945*	*M0ZS78*	*PK1*	INT	Pyruvate kinase 1	[1]
*PGSC0003DMT400002870*	*M0ZKT2*	*H3.2*	DOMc	Histone H3.2-like	[19,20,21,22,23,24,25,26,27,28,29,30]
*PGSC0003DMT400029242*	*M1ASG7*	*GAPCP1*	INT	Glyceraldehyde-3-phosphate dehydrogenase	[31,32,33,34,35,36,37,38,39,40,41,42,43]
*PGSC0003DMT400032266*	*M1AX44*	*MMDH*	INT	Malate dehydrogenase	[34]
*PGSC0003DMT400076602*	*M1CX22*	*FPS1*		Farnesyl pyrophosphate synthase 1-like	[18,44,45,46,47,48,49,50,51,52,53,54,55]
*PGSC0003DMT400071330*	*M1CNK1*	*TPI*	DOMc	Triosephosphate isomerase	[56,57,58,59,60,61]
*PGSC0003DMT400071725*	*M1CP75*	*RPL4*	DOMc	60S ribosomal protein L4	[62,63,64,65,66,67]
*PGSC0003DMT400070920*	*M1CMY9*	*SOD/Fe*		Superoxide dismutase [Fe]	[68,69,70,71,72,73,74,75]
*PGSC0003DMT400007585*	*M0ZT85*	*KAS2*		3-oxoacyl-[acyl-carrier-protein] synthase II	[76,77]
*PGSC0003DMT400062986*	*M1C9 × 0*	*ENO1*	DOMc	Enolase 1	[78,79,80]
*PGSC0003DMT400077358*	*M1CYA5*	*HSP70-8*	INT	Heat shock 70 kDa protein 8	[81,82,83,84]
*PGSC0003DMT400035521*	*M1B2E4*	*PER*	INT	Peroxidase	[85,86,87,88,89,90,91,92,93,94,95,96]
*PGSC0003DMT400001937*	*M0ZJD1*	*THY-1*		Bifunctional dihydrofolate reductase-thymidylate synthase-like	[97]
*PGSC0003DMT400056871*	*M1C005*	*PGK*		Phosphoglycerate kinase	[98]
*PGSC0003DMT400087679*	*Q38JH8*	*METK2*		S-adenosylmethionine synthase 2	[99]
*PGSC0003DMT400060739*	*M1C6C4*	*RPL51*		54S ribosomal protein L51	[100]
*PGSC0003DMT400036981*	*M1B4L2*	*ACP*		Acyl carrier protein 1	[101]

**Table 3 plants-13-02996-t003:** Primer set of stress resistance DEGs used to validate their expression levels.

Potato ID String v11.5	ID String v.12	ID	NCBI	Forward	Reverse
*PGSC0003DMT400001937*	*M0ZJD1*	*THY-1*	XM_015304289.1	GTGCTAAGGTCCTACAAGGAAG	CCAAATCACCCTCTTCCCTATC
*PGSC0003DMT400002870*	*M0ZKT2*	*H3.2*	XM_006349079.2	GTATCAGAAGTCGACGGAGTTG	ACCTCAGATCCGTCTTGAAATC
*PGSC0003DMT400006945*	*M0ZS78*	*PK1*	XM_006341124.2	CGAAGAGGGCTTGACACATT	CCTTCTCAGGTGGGAGATCTAT
*PGSC0003DMT400076602*	*M1CX22*	*FPS1*	XM_006344841.2	GGAGGTGTACTCTGTGCTTAAA	GATAGTCCTCGATTCAGCTTCC
*PGSC0003DMT400029242*	*M1ASG7*	*GAPCP1*	XM_006352526.2	GGTTACACAGACGAGGATGTT	GAGACGAGCTTCACGAATGA
*PGSC0003DMT400056871*	*M1C005*	*PGK*	NM_001288522.1	CCACTTGTGCCTAGACTTTCA	AGTTCAGCCACCAAGTTCTC
*PGSC0003DMT400062986*	*M1C9 × 0*	*ENO1*	XM_006345012.2	CTACTAACGTCTCCTCCAAAGC	ATAGGAAAGTCCGCCGAAAG
*PGSC0003DMT400071725*	*M1CP75*	*RPL4*	XM_049512576.1	TGAGGCACAAAGAGTCAAGG	TTTGCCTGCTGACCTGATAG
*PGSC0003DMT400087679*	*Q38JH8*	*METK2*	NM_001318549.1	GACTTGCTCGTCGCTGTATT	TCGGGAATTGTTCCTGTCTTG
*PGSC0003DMT400032266*	*M1AX44*	*MMDH*	NM_001288105.1	CGCACCAGAGAGGAAAGTT	CGTAGAGTGAAAGGCTGGTAC
*PGSC0003DMT400071330*	*M1CNK1*	*TPI*	NM_001318582.1	TGGGCTATTGGTACTGGAAAG	GCAGCAACTTCAGCACTAAC
*PGSC0003DMT400035521*	*M1B2E4*	*PER*	XM_006364783.2	GCACAGTTTCCAACGCTAAAG	GGACAACCAAGTCGAGAACA
*PGSC0003DMT400070920*	*M1CMY9*	*SOD/Fe*	XM_006357250.2	GGCCTGGAATCATCAGTTCTT	GCTGCAGCTGCCTTAAATTC
*PGSC0003DMT400060739*	*M1C6C4*	*RPL51*	XM_006346690.2	GCCGACGTTCTACTTCTTACTC	TAGCTGACTACCAGCTTCCT
*PGSC0003DMT400007585*	*M0ZT85*	*KAS2*	XM_006345186.2	GGTGGATCAGAGGCAGTAATTG	CAAGGCCGAGAAGCTTTAGTAG
*PGSC0003DMT400036981*	*M1B4L2*	*ACP*	XM_006341431.2	ACATCCCGCTTTCGTGTT	GTACTTTCAGGGCTGACCTTAG
*PGSC0003DMT400077358*	*M1CYA5*	*HSP70-8*	XM_006360745.2	GCTCGTCAGAAACACGAGAA	CGTGAGCCAGTTCATCACTAA

## Data Availability

The raw data supporting the conclusions of this article will be made available by the authors on request.

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
