# Peer review of "Analysis of Stress Response Genes in Microtuberization of Potato Solanum tuberosum L.: Contributions to Osmotic and Combined Abiotic Stress Tolerance"

_plants, 2024, doi:10.3390/plants13212996_

Round 1
Reviewer 1 Report
Comments and Suggestions for Authors
The article “Analysis of stress response genes in microtuberization of Solanum tuberosum: contributions into osmotic, heat, cold, salt, and combined stress tolerance” addresses the question “What are the molecular mechanisms and specific genes involved in potato (Solanum tuberosum) microtuberization under different abiotic stress conditions, and how can these genes be targeted to enhance stress tolerance?” The study aims to identify key genes involved in osmotic, heat, cold, salt, and combined stress tolerance to improve the resilience of potato crops through transcriptomic and protein-protein interaction (PPI) analyses.
This research is relevant to the field of plant stress physiology and molecular biology, specifically focusing on improving crop resilience to abiotic stresses, which are a major challenge to global food security. The specific gap addressed by the paper is the limited understanding of the key genes and molecular networks involved in microtuber development under various stress conditions in potato. While previous studies have focused on individual stresses, this paper addresses combined stresses, providing insights into how potato plants manage complex stress environments through gene regulation and expression.
The article contributes to the growing body of knowledge on the molecular basis of stress tolerance in potato, distinguishing itself by exploring the interaction between multiple abiotic stress factors (osmotic, heat, cold, salt, and combined stress). By identifying specific ancestral and introgressed genes that play crucial roles in stress responses, the study advances the understanding of how these genes contribute to the adaptability and survival of potatoes under challenging conditions.
The methodology is robust, however, should be interesting to perform gene knockout/overexpression experiments to confirm the function of the identified genes. This would strengthen the claims about gene function and its potential for improving stress tolerance.
The conclusions drawn by the authors are generally consistent with the evidence presented. However, one area that could be more thoroughly addressed is the specific role of cold stress, as the study suggests that the gene H3.2 does not contain motifs directly related to cold stress, despite being highlighted in the cold stress response. More detailed experimental validation would strengthen this claim. Additionally, the authors could clarify whether all the main research questions posed were addressed by the specific experiments conducted, such as gene interactions under combined stresses and the exact functional role of identified genes in microtuber formation.
The references cited are appropriate and include a range of recent studies on plant stress physiology, molecular biology, and potato genetics.
Overall, this paper provides a significant contribution to the field, addressing a key gap in the understanding of potato stress tolerance and offering practical implications for future crop improvement strategies. With some methodological refinements, the study could provide even more valuable insights into the molecular mechanisms underpinning stress resilience in potatoes.
Author Response
Dear Reviewer 1, we appreciate your kind comments. This motivate us to follow the improvement of the understanding of the molecular mechanisms involved in this changing climate conditions.
The genome edition by up and down regulation of genes will help us to elucidate this problems
Reviewer 2 Report
Comments and Suggestions for Authors
The effects of osmosis, heat, cold and salt stress on potato microtuber development were compared, and the associated genes were screened for comprehensive analysis. The research content has certain academic value and practical significance. However, there are still some key issues that are not clearly stated, and the author should explain them in the following aspects.
Line 132: Are the genes in Table 2 up-regulated under all four stresses? On what basis were they chosen? How much did they up-regulate in RNA-Seq?
Line 186: qRT-PCR analysis should be performed using the mainstream 2-ΔΔCt method.
Line 511: What method was used for transcriptome analysis? How many groups and which samples were used for transcriptome analysis? Why is the basic transcriptome data not presented in the results?
Line 532: 2-ΔΔCt analysis is used in the method, but log2 is shown in Figure 5. Why?
Line 545: Why the NCBI number is used in Table 3, it should be the same as in Table 2.
Line 565: what is "Patents"?
Author Response
Reviewer 2:
The effects of osmosis, heat, cold and salt stress on potato microtuber development were compared, and the associated genes were screened for comprehensive analysis. The research content has certain academic value and practical significance. However, there are still some key issues that are not clearly stated, and the author should explain them in the following aspects.
Dear reviewer 2, we have answered your interesting questions about this work. We appreciate your time and effort taken tho the manuscript.
Are the genes in Table 2 up-regulated under all four stresses?
Dear reviewer 2, the answer is No, the genes enlisted in Table 2 were upregulated in osmotic stress conditions revealed in the transcriptome analysis previously published (Valencia-Lozano 2022).
On what basis were they chosen?
We made a literature analysis and selected genes that were involved in heat (10), cold (11), salt (17) and osmotic stress (12) that were interacting with cell cycle, sterol, ribosomal proteins, nucleosome and carbon metabolism.
The PPI network devised in STRING database revealed a set of 343 upregulated genes under osmotic stress. The PPI network consists in essential modules for life; cell cycle, ribosomal proteins, nucleosome interacting with carbon metabolism, sterol biosynthesis, fatty acid biosynthesis and metabolism.
How much did they up-regulate in RNA-Seq?
|
Potato ID String v11.5 |
A. thaliana |
RNA_Seq Log2 |
Annotation |
|
PGSC0003DMT400006945 |
PK1 |
1.058104124 |
Pyruvate kinase 1 |
|
PGSC0003DMT400029242 |
GAPCP1 |
1.42008818 |
Glyceraldehyde-3-phosphate dehydrogenase |
|
PGSC0003DMT400032266 |
MMDH |
1.003011916 |
Malate dehydrogenase |
|
PGSC0003DMT400035521 |
PER |
1.959469831 |
Peroxidase |
|
PGSC0003DMT400077358 |
HSP70-8 |
1.06094563 |
Heat shock 70 kDa protein 8 |
|
PGSC0003DMT400002870 |
H3.2 |
1.965740804 |
Histone H3.2-like |
|
PGSC0003DMT400062986 |
ENO1 |
1.157499906 |
Enolase 1 |
|
PGSC0003DMT400071725 |
RPL4 |
1.993851219 |
60S ribosomal protein L4 |
|
PGSC0003DMT400071330 |
TPI |
1.0312709 |
Triosephosphate isomerase |
|
PGSC0003DMT400001937 |
THY-1 |
1.069524705 |
Bifunctional dihydrofolate reductase-thymidylate synthase-like |
|
PGSC0003DMT400076602 |
FPS1 |
1.127147537 |
Farnesyl pyrophosphate synthase 1-like |
|
PGSC0003DMT400056871 |
PGK |
1.238227853 |
Phosphoglycerate kinase |
|
PGSC0003DMT400087679 |
METK2 |
1.615304632 |
S-adenosylmethionine synthase 2 |
|
PGSC0003DMT400070920 |
SOD/Fe |
1.202185043 |
Superoxide dismutase [Fe] |
|
PGSC0003DMT400060739 |
RPL51 |
1.442026192 |
54S ribosomal protein L51 |
|
PGSC0003DMT400007585 |
KAS2 |
1.570944696 |
3-oxoacyl-[acyl-carrier-protein] synthase II |
|
PGSC0003DMT400036981 |
ACP |
1.657153353 |
Acyl carrier protein 1 |
Line 186: qRT-PCR analysis should be performed using the mainstream 2-ΔΔCt method.
Dear Reviewer 2, indeed, the qRT-PCR analyses were interpreted using the 2-ΔΔCt method. Relative expression estimation levels are represented in Log2-Fold Change. It was made according to Livak et al., 2001, by using two endogenous genes EF1 and SEC3, validated by triplicate in each treatment.
Line 511: What method was used for transcriptome analysis?
The methodology was described according to Valencia-Lozano et al 2022:
Valencia-Lozano, E., Herrera-Isidrón, L., Flores-López, J. A., Recoder-Meléndez, O. S., Barraza, A., & Cabrera-Ponce, J. L. (2022). Solanum tuberosum Microtuber Development under Darkness Unveiled through RNAseq Transcriptomic Analysis. International Journal of Molecular Sciences, 23(22), 13835.
Transcriptome Sequencing and Assembly Sequenced reads were tested for quality using the FastQC software package (http://www.bioinformatics.babraham.ac.uk/projects/fastqc/) and preprocessed to remove sequence adapters and low-quality bases using the software Trimmomatics (Adapter removal was performed using the Trimmomatic v0.3.6 program. RNA-seq reads were aligned to the Solanum tuberosum reference genome available in Phytozome v12.1. and Solgenomics with the STAR aligner v.2.5.2b. In this step, the BAM (Binary Alignment/Map) files were generated. Subsequently, a count and set of transcripts were made using the featureCounts program of the Subread v.1.5.2 package). A quantification and differential analysis of the transcripts was performed using the DESeq2 v1.12.4 program. Finally, an ontology analysis was performed using Blast2GO.
How many groups and which samples were used for transcriptome analysis?
There were two groups, the first is explants producing microtubers after 15 days of incubation in MR8G6-2iP under darkness and the second one, explants under non-microtuber medium, MR1G3-2iP under darkness conditions. In both groups, triplicate samples was used.
Why is the basic transcriptome data not presented in the results?
In this study, the transcriptome was not performed; rather, the transcriptome data from Valencia-Lozano et al. 2022 was used to construct the protein-protein interaction (PPI) network and to select the genes of interest in the different clusters.
Line 532: 2-ΔΔCt analysis is used in the method, but log2 is shown in Figure 5. Why?
Log2 transformation is often applied to the results of the 2-ΔΔCt analysis to make the data easier to interpret and visualize. This is because gene expression changes are typically multiplicative, and the log2 scale converts these multiplicative changes into additive ones, which are easier to understand and plot on graphs
Line 545: Why the NCBI number is used in Table 3, it should be the same as in Table 2.
Because the NCBI number is an identification of the nucleotide sequence of NCBI database. As you can see, in table 2, it is not shown this number.
Line 565: what is "Patents"?
No patent were applied in this work.

Round 2
Reviewer 2 Report
Comments and Suggestions for Authors
The author's revisions to the manuscript have improved the quality of the manuscript to some extent, but there are some problems that authors seems to have failed to understand the true meaning. The authors needs to look at these issues comprehensively and make revisions in the manuscript.
Line135: The number of genes responding to stress is huge, Table 2 only shows 17 of them, why they were selected, it is not enough to answer the reviewer's opinions, the author also needs to record in detail in the manuscript, so that readers can understand. Moreover, the data of RNA-seq provided by the authors are all less than 2.0, and their upregulation seems not obvious, so what is the significance of selecting them?
Line147: PGSC
Line525: Since transcriptome sequencing was not done, why write transcriptome sequencing in the title? If data from other scholars is used, it should be explicitly cited in the manuscript.
Line560: Some genes in Table 2 and Table 3 are the same, and they must use the same gene ID.
Line580: Papers generally do not need to state whether they are patent or not, please read the journal manual carefully.
Author Response
The author's revisions to the manuscript have improved the quality of the manuscript to some extent, but there are some problems that authors seems to have failed to understand the true meaning. The authors needs to look at these issues comprehensively and make revisions in the manuscript.
Dear Reviewer/Editor, thank you very much for your observations about our manuscript. We have modified the manuscript according to your suggestions.
1.- The PPI network 2.2 was deleted.
2.- We have explained point by point the selection of genes interacting with the respective modules.
2.2. Identification of stress-responsive genes
Two criteria were used to select stress responsive genes involved in MTs development analysis: i) the genes interacting with the essential life and developmental modules, such as nucleosome, cell cycle and ribosomal proteins previously reported by Valencia-Lozano et al. 2022, 2023 [16, 17], Herrera-Isidrón et al. 2024 [18] (Figure 2); and ii) a systematic review of published reports based on functional mutants and the up- and downregulation analysis focused in to osmotic, heat, cold, salt, and combined stress analysis.
Then, 17 stress-response genes were selected (Table 2). Those genes interact with cell cycle and carbon metabolism, such as the bifunctional dihydrofolate-reductase-thymidylate synthase-like (PGSC0003DMT400076602), S-adenosylmethionine synthase 2 (PGSC0003DMT400087679), Heat shock 70 kDa protein 8 (PGSC0003DMT400077358); genes interacting with ribosomal proteins, such as 60S ribosomal protein L4 (PGSC0003DMT400071725M1CP75), 54S ribosomal protein L51 (PGSC0003DMT400060739), Superoxide dismutase [Fe] (PGSC0003DMT400070920), Peroxidase (PGSC0003DMT400035521); a gene of the nucleosome module interacting with carbon metabolism: Histone H3.2-like, (PGSC0003DMT400002870); a gene of carbon metabolism interacting with nucleosome: Glyceraldehyde-3-phosphate dehydrogenase, (PGSC0003DMT400029242); genes involved in carbon metabolism, such as Pyruvate kinase 1 (PGSC0003DMT400006945), Malate dehydrogenase, (PGSC0003DMT400032266), Enolase 1 (PGSC0003DMT400062986), Phosphoglycerate kinase (PGSC0003DMT40005687), Triosephosphate isomerase (PGSC0003DMT400071330); genes involved in fatty acid metabolism interacting with cell cycle, such as 3-oxoacyl-[acyl-carrier-protein] synthase II, (PGSC0003DMT400007585) and Acyl carrier protein 1 (PGSC0003DMT400036981); and a gene interacting with sterol biosynthesis: Farnesyl pyrophosphate synthase 1-like, PGSC0003DMT400076602 (Table 2, Figure 2).
From the 17 selected genes, nine genes were part of the list published by Hardigan et al. 2017, supplementary data S4 and S9 [1]. Of these nine selected genes, five genes were corresponded to introgressed (INT) genes, and four were to domesticate (DOMc) genes. And the nine genes were identified in both landraces and cultivars (Table 2) [1]. The selected INT genes were NAD-malate dehydrogenase (PGSC0003DMG400019511), Glyceraldehyde-3-phosphate dehydrogenase (PGSC0003DMG400004130), Class III peroxidase (PGSC0003DMG400006993), Heat shock protein 70-8 (PGSC0003DMG400014835), and Pyruvate kinase (PGSC0003DMG400024220) (Table 2) [1]. The DOMc were regarded as genes under selection in both landraces and cultivars, with an impact on performance regardless of hemispheric or local adaptation. The genes were Histone H3.2 (PGSC0003DMG400001119), chloroplastic Triosephosphate isomerase (PGSC0003DMG400004436), Enolase1 (PGSC0003DMG400011044), chloroplastic Ribo-somal protein L4 (PGSC0003DMG400007051) (Table 2) [1].
3.- In the table 2 we added references from the systematic review.
Line135: The number of genes responding to stress is huge, Table 2 only shows 17 of them, why they were selected, it is not enough to answer the reviewer's opinions, the author also needs to record in detail in the manuscript, so that readers can understand.
Dear Reviewer We have modified the explanation within the manuscript about the criteria we have used in this manuscript.
Moreover, the data of RNA-seq provided by the authors are all less than 2.0, and their upregulation seems not obvious, so what is the significance of selecting them?
Answer:
For transcriptomic analysis through RNA-seq, the standard threshold value applied to take as significant increase or decrease expression levels compared against the control conditions is above the absolute value of 2-fold change (|>2|), that is log2, 2 = 1 or 2^1 = 2 (Love, et al. 2014). Also, this criterion is applied for quantitative PCR transcriptional level analysis (Livak & Schmittgen, 2001).
Livak KJ, Schmittgen TD. Analysis of relative gene expression data using real-time quantitative PCR and the 2(-Delta Delta C(T)) Method. Methods. 2001 Dec;25(4):402-8. doi: 10.1006/meth.2001.1262.
Line147: PGSC
Line525: Since transcriptome sequencing was not done, why write transcriptome sequencing in the title?
Answer: The title of this manuscript does not contain any single word about Transcriptome sequencing, the title is:
Analysis of stress response genes in microtuberization of Solanum tuberosum: contributions into osmotic, heat, cold, salt, and combined stress tolerance.
Lisset Herrera-Isidron1,†, Braulio Uribe-Lopez1,†, Aaron Barraza3, José Luis Cabrera-Ponce,*, Eliana Valencia-Lozano,*
If data from other scholars is used, it should be explicitly cited in the manuscript.
Line560: Some genes in Table 2 and Table 3 are the same, and they must use the same gene ID.
Answer:
We have added the ID information in table 3.
Line580: Papers generally do not need to state whether they are patent or not, please read the journal manual carefully.
Answer:
Thank you very much for your observation, we deleted from the manuscript this part.
Dear editor I hope the edition of our manuscript according to your suggestions will fulfill the requirements for publication in Plants mdpi Journal.
Yours
Dr. José Luis Cabrera Ponce

Round 3
Reviewer 2 Report
Comments and Suggestions for Authors
After the author's revision, the quality of the manuscript has been improved. Please pay attention to some writing details to make it better.
1. The names of species in Figure 3 are too small to read clearly.
2. What does the box in front of the number in Figure 6A mean?
3. The words in Figure 8 are not clear, such as "osmosis".
4. Line 336-339: There is only one sentence here, does it need to be a separate topic?
Author Response
After the author's revision, the quality of the manuscript has been improved. Please pay attention to some writing details to make it better.
- The names of species in Figure 3 are too small to read clearly.
Dear reviewer, it was corrected.
- What does the box in front of the number in Figure 6A mean?
Dear reviewer, the box A and box B correspond to the next explanation:
Figure 6. A: Principal component analysis (PCA) from the relative gene expression under different type of stresses during microtuberization of potato under darkness. The group of genes with higher variance were H3.2, and GAPCP1 involved in osmotic, cold, and heat stress. Combined-stress was associated with FPS1, TPI, RPL4 and SOD/Fe, and salt stress was associated with KAS2, ENO1, HSP70-8, and PER. B: Corr PCA, Dimension 1 (PC1) shows a uniform distri-bution in the number of genes, indicating a general variability in gene expression without bias toward a specific treatment. In contrast, Dimension 2 (PC2) reveals a high load of genes associat-ed with osmotic stress response, highlighting their predominant relevance in this dimension.
- The words in Figure 8 are not clear, such as "osmosis".
Dear reviewer the figure 8 was corrected
- Line 336-339: There is only one sentence here, does it need to be a separate topic?
Dear reviewer, totally agree, the sentence was eliminated
